# Image-based cell profiling enhancement via data cleaning methods

**Arghavan Rezvani**◉⊛, **Mahtab Bigverdi**◉⊛, **Mohammad Hossein Rohban**◉*

Department of Computer Engineering, Sharif University of Technology, Tehran, Tehran, Iran

⊛ These authors contributed equally to this work.
* rohabn@sharif.edu

**Data Availability Statement:** URL: http://gigadb.org/dataset/view/id/100351 DOI:10.5524/100351.

**Funding:** The author(s) received no specific funding for this work.

## Abstract

With the advent of high-throughput assays, a large number of biological experiments can be carried out. Image-based assays are among the most accessible and inexpensive technologies for this purpose. Indeed, these assays have proved to be effective in characterizing unknown functions of genes and small molecules. Image analysis pipelines have a pivotal role in translating raw images that are captured in such assays into useful and compact representation, also known as measurements. CellProfiler is a popular and commonly used tool for this purpose through providing readily available modules for the cell/nuclei segmentation, and making various measurements, or features, for each cell/nuclei. Single cell features are then aggregated for each treatment replica to form treatment "profiles". However, there may be several sources of error in the CellProfiler quantification pipeline that affects the downstream analysis that is performed on the profiles. In this work, we examined various preprocessing approaches to improve the profiles. We consider the identification of drug mechanisms of action as the downstream task to evaluate such preprocessing approaches. Our enhancement steps mainly consist of data cleaning, cell level outlier detection, toxic drug detection, and regressing out the cell area from all other features, as many of them are widely affected by the cell area. Our experiments indicate that by performing these time-efficient preprocessing steps, image-based profiles can preserve more meaningful information compared to raw profiles. In the end, we also suggest possible avenues for future research.

## Introduction

High-throughput image-based assays have proved to be an effective predictive tool in the early stages of drug discovery through automated microscopy and image analysis, which make quantification of cellular morphological responses possible at a large scale [1]. These experiments often involve growing cells in multi-well plates and then treating cells in each well with a small molecule, or genetic perturbation. Image-based profiling has diverse and powerful applications, including identification of gene and allele functions and targets, and mechanisms of action (MoA) of drugs [2]. Prediction of drug MoAs through such assays potentially saves drug discovery process costs when applied early on. According to previous studies, the MoA of

**Competing interests:** The authors have declared that no competing interests exist.

unknown compounds can be predicted by grouping each unknown compound with already-annotated compounds based on the similarity of their morphological profiles [3–5]. Therefore, high throughput assays can be helpful in this process.

The typical workflow in the analysis of images that are produced by high-throughput assays includes illumination correction, nuclei/cell segmentation, quality control, morphological feature measurement, batch effect removal, data normalization, feature selection/dimensionality reduction, and finally, aggregation of single cell measurements into image-based profiles per well [2]. The initial steps such as the illumination correction, segmentation, and feature extraction are not investigated in this work. We instead mainly focus on approaches that preprocess the features extracted by CellProfiler, open-source software that aims to automate most of these steps. We conduct a comprehensive study on how such techniques could potentially improve the profile's quality. Data cleaning is a key step for enhancing image-based profiling as there may be different artifacts in the staining and imaging process and can affect the next steps. There are some quality control methods applied to raw images; for instance, in [6], a cell-level quality control approach has been proposed based on the high throughput images. However, examining raw images can be time-consuming and not applicable to previously prepared datasets. Hence, in this work, we focus on extracted image-based profiles as the input to the downstream analysis. A significant step in image-based profiling and data cleaning is cell-level quality control. Outlier cells, which do not show any valid biological effect, may result from errors in different parts of the pipeline. For instance, an error in the segmentation step may result in overly small or large cells and bias the profiles heavily as a result. Detecting and removing outlier cells can highly improve the profile quality.

We can categorize strategies for detecting outlier cells into two categories [2]: model-free and model-based. In model-free strategies, statistical analysis is used to detect outlier cells; for instance, calculating estimators such as median and median absolute deviation for multivariate situations is useful [7]. The principal component analysis is another model-free strategy for outlier detection [8]. In model-based strategies, a model is trained based on normal samples. It can be a linear regression model, in which outliers can be detected as data points with a large residual [9]. Alternatively, by providing samples of outliers, supervised machine learning classifiers can be beneficial [10].

We propose to use a unsupervised outlier detection method in this work and evaluate its effectiveness in the context of MoA prediction. More specifically, histogram-based outlier detection is applied on each plate individually, and by removing the single-cells detected as outliers by this method, final results are improved.

In addition, we found that the domain-specific feature preprocessing, as a data cleaning step, could enhance the relevant downstream task. In summary, we found that the cell area is a pivotal contributor to many of the features, and hence the similarity metric is highly influenced by it. Therefore, we propose to neutralize the effect of cell area on the other features to be able to capture more meaningful information in the profiles. The major steps of this work are described in Fig 1. Finally, we would investigate various general purpose methods for dimensionality reduction and representation learning from the classical features that are extracted by CellProfiler, and found that, as opposed to the domain specific preprocessing step, they are not effective in improving the prediction ability of profiles.

According to previous studies with the same metric as the proposed method of this work, increasing the enrichment of profiles is not an easy task in the context of high-throughput microscopy [11]. However, we suggest some efficient preprocessing methods that can increase the metric significantly, which is our main contribution.

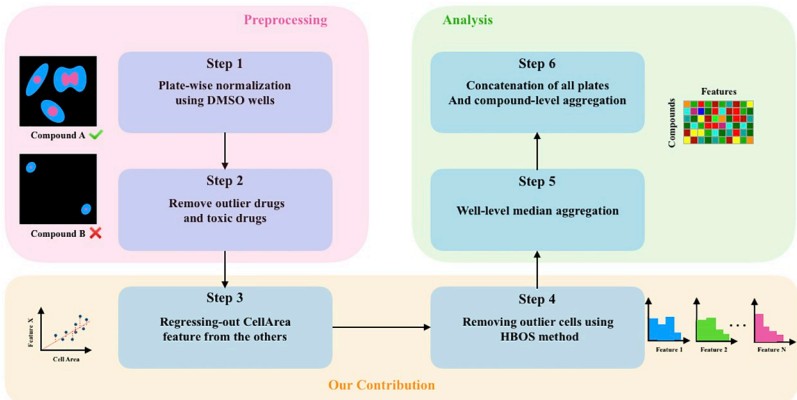

**Fig 1. Representative workflow for the enhanced image-based cell profiling via data cleaning.** Six main steps transform cell measurements of each plate to compound-level profiles. Specifically, in step 3, we replace each feature by part of those features that are accounted for by the cell area feature. This is achieved by regressing each feature against the cell area, and replacing each feature by the residue of the corresponding regression. Furthermore, cells that appear as outliers are identified and removed through a histogram-based approach, which is called HBOS, to make sure that the well-level aggregated profiles would not be affected by potential artifacts.

## Materials and methods

### Dataset

We used a dataset of images and morphological profiles of small-molecule treatments using the Cell Painting assay [12]. About 50 plates, which correspond to the bioactive compounds with known mechanisms of action, are chosen. Considering redundant features extracted by CellProfiler, a specific list of features is selected for this experiment. This feature and plate selection is based on the method proposed by Rohban et al. [11]. Each plate consists of wells that consist of cells, and each well is treated by a particular compound. There are wells known as negative controls (DMSO), which are treated only with the compound solvent. The profile of a well is obtained by taking the median of single-cell measurements in that well. To mitigate the batch effect, profiles in each plate were normalized separately using the DMSO wells of that plate. For normalizing cells of a plate, single cells were subtracted by the median of DMSO well profiles and divided by the median absolute deviation (MAD) of DMSO well profile [4]. There are several strategies for creating aggregated profiles. In this work, similar to earlier experiments [4, 11], the median and profiles are used for well-level and profile-level aggregations.

### Methodology

**Preprocessing.** There are several sources of error while extracting raw image-based single-cell profiles that may dramatically influence further steps in the pipeline. Consequently, preprocessing strategies are necessary to get more meaningful and stable results. For instance, some of the wells that contained DMSO had some intensity issues. This may mislead CellProfiler algorithms and cause them to extract noisy information. Noisy DMSO can affect all the information of a plate, as it is used for normalization. In Figs 2 and 3, images of some misleading DMSO wells can be seen.

Some wells had an extraordinary high cell area. By investigating their raw images, it was clear that it was CellProfiler's mistake. Those images mostly were toxic wells containing a very

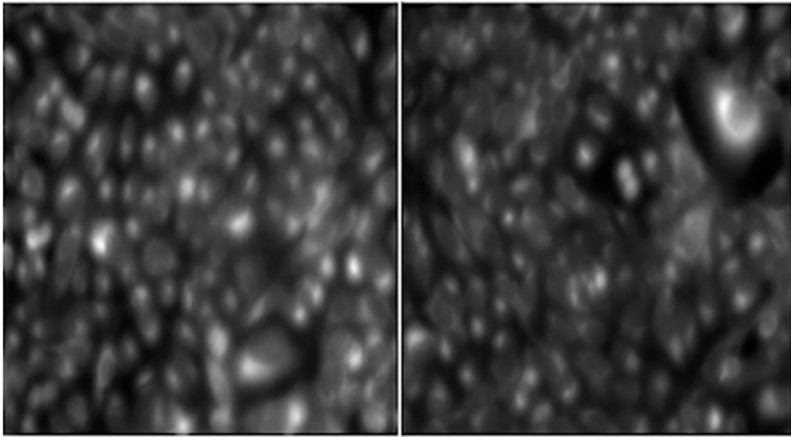

**Fig 2. Plate 24277, well A13, containing DMSO, is blurry.**

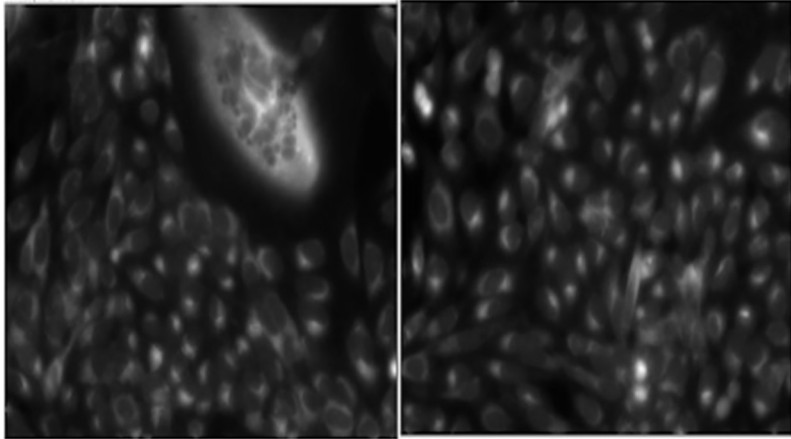

**Fig 3. Another outlier DMSO well in plate 24293, which shows a huge clump on the upper right corner of the left image.**

small number of cells. For example, plate 24293, well E19, has a cell area of nearly 3000 after normalization (Fig 4).

The noise in the images, especially in DMSO wells, could be a source of error in the Cell-Profiler pipeline, and this indicated the importance of data cleaning in image-based profiling enhancement. Preprocessing steps remove drugs with unrecognizable effects on the features (outlier drugs) as well as the toxic drugs.

**Removing outlier drugs.** Not all compounds produce a meaningful change in the features extracted by CellProfiler. Therefore, a drug selection method must be used to only keep those drugs with discernible effects on the features [13]. First, a median profile for each well is calculated. Then, the wells are grouped based on their "Metadata_broad_sample" column, so each group contains profiles of a specific compound. The similarity measure is chosen to be Pearson's correlation, which reflects the linear relationship between two variables. The more similar the profiles are between two wells, the higher the correlation coefficient will be. To select drugs with meaningful effect on the features, a null distribution is required. The null

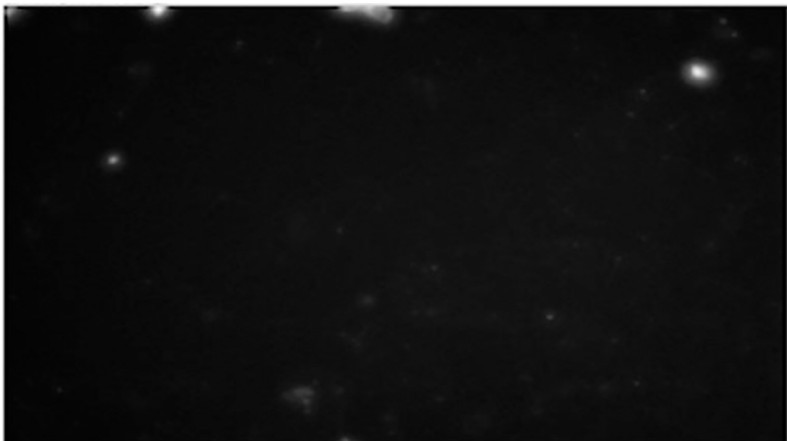

**Fig 4. Plate 24293, well E19, with a cell area of nearly 3000.**

distribution is defined based on the correlations between non-replicates. That is, the median correlation between groups of k different compounds constitute samples of the null distribution, where k is the number of technical replicates in the experiment (k = 3 in this work). Compounds whose median replicate correlations are greater than the 95th percentile of the null distribution are selected for further steps [13, 14], as they have a meaningful effect on the cells. We will refer to the drugs removed in this section as the "outlier drugs" in the following sections. In this experiment, there were a total of 1551 different compounds, and based on the explained method, only 455 of them were kept.

**Removing toxic drugs.** While investigating the features of each well, we faced some wells with an extraordinarily low number of cells, even less than 10 cells in a whole well, as shown in Fig 5. This issue could have several sources. It could be the result of a bad vital situation such

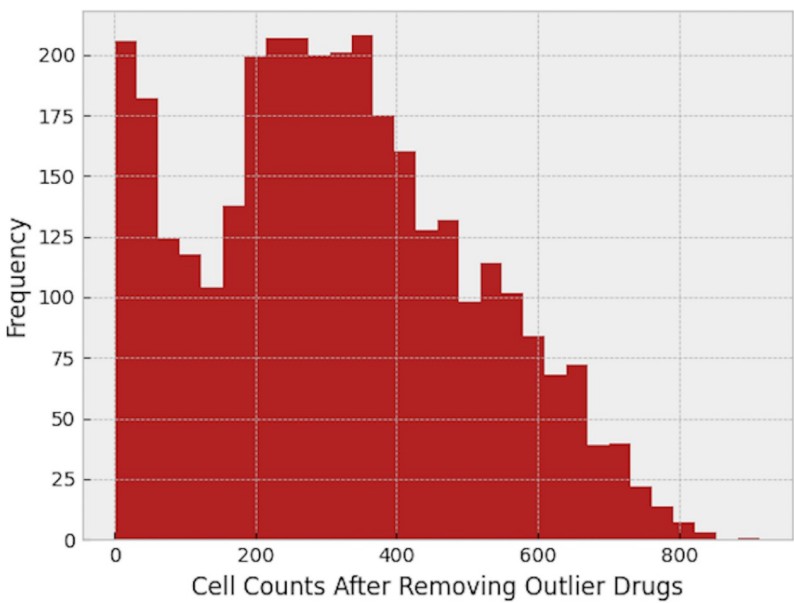

**Fig 5. Histogram of the cell count in each well.**

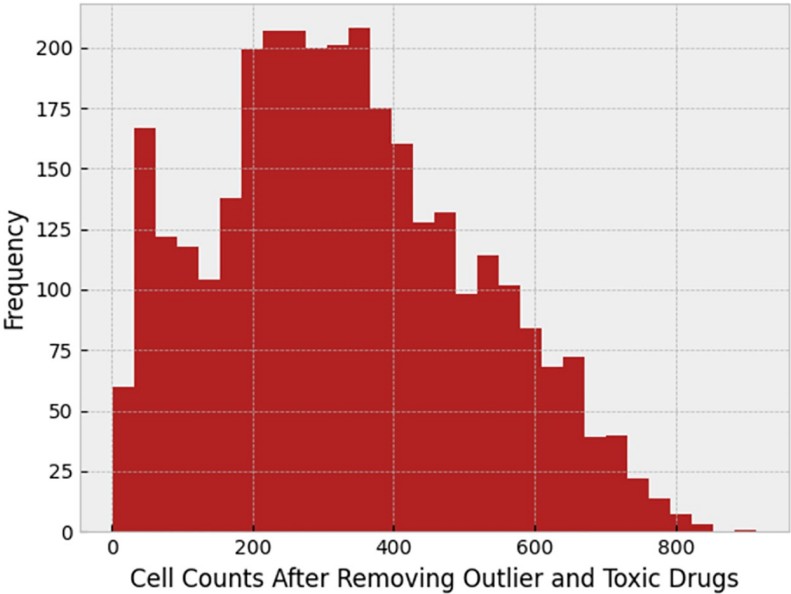

**Fig 6. Histogram of the cell count after removing toxic drugs.**

as lack of nutrients in a well, which caused the cell death. Or, it could be the result of the drug itself. Some drugs have a toxic nature and cause cell death.

If the drug is toxic, that may cause a false result in the final odds ratio plot. To find toxic drugs, we checked the median of cell count of different wells facing the same drug, and we removed 0.05 of the drugs with the lowest cell count median. Cell count histogram after removing toxic drugs can be found in Fig 6.

After doing so, the odds ratio was decreased. This indicates that the toxic drugs have a correlation in their features that by removing them from the experiment, odds ratio decreases, and toxic drug removal makes the final result more real. Fig 7 shows the correlation between cells affected by toxic drugs, and Fig 8 corresponds to the correlation between toxic and non-toxic drugs. The comparison between these figures clarifies that toxic drugs are highly correlated.

**Evaluation metric.** Since similarity metrics reveal connections among image-based profiles, choosing a suitable one can enhance the analysis. Like similar tasks, Pearson's correlation coefficient was used in this work.

Consider two wells, each treated by a compound, which has some special mechanism of action, or "MoA" for short. In this experiment, the final purpose is to find whether there is any association between high values of correlations of the profiles of two wells and similarity of the MoA of compounds that are used to treat the cells in those wells. In other words, it is hypothesized that if profiles of two compounds have a high correlation, they probably have the same MoA. To test this hypothesis, one-sided Fisher's exact test is used.

Suppose Fisher's test admits the hypothesis. In that case, this process can be used for drug discovery, as it is possible to do high throughput experiments and use the correlation of profiles to guess the MoA of unknown compounds. In this experiment, odds ratio is used to check how likely it is that the hypothesis is true. The Fisher's test contingency table used in our experiment can be found in Fig 9.

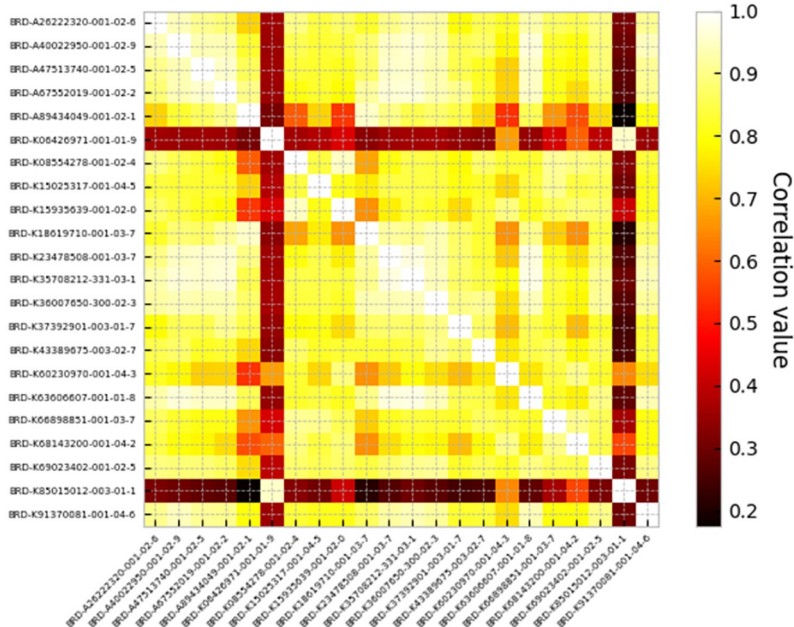

**Fig 7. Correlation between toxic drugs.**

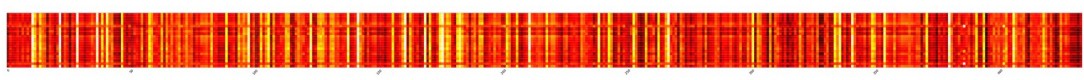

**Fig 8. Correlation between toxic drugs and non-toxic drugs.** Toxic drugs are in rows and non-toxic ones are in columns.

We change the value of "k" in Fig 9 to draw a plot, where its y axis corresponds to the odds ratio value, and its x axis corresponds to value "k", or the percentage of pairs with the highest correlations.

If the hypothesis is true, we expect to have a plot with a negative slope. That means that by tightening the condition of correlation, there must be more similarities in MoAs.

**Histogram-based outlier score.** By investigating images of some wells and the histogram of multiple features, it was crystal clear that there are some outlier profiles. The cause of this could be the issues in the pipeline or the imaging process. There are several ways to detect outliers in a dataset. Three different setups for outlier detection are supervised, semi-supervised, and unsupervised methods. For the first two setups, labeled data is needed [15]. In this dataset, outlier cells are not labeled, therefore unsupervised methods have to be adopted. Outputs of outlier detection methods can be scores or labels. Labels (normal or outlier) are assigned to each instance individually.

There are various approaches for unsupervised outlier/anomaly detection. Well-known methods are Local Outlier Factor (LOF) [16], nearest-neighbor-based algorithm [17], Clustering Based Local Outlier Factor (CBLOF) [18], and LDCOF [19]. These are computationally expensive. A fast and efficient unsupervised method for outlier detection is the Histogram-

|  | Same MoA | Different MoAs |
|---|---|---|
| Compound pairs with highest correlation (*k* percent of all pairs) | A | B |
| Other pairs | C | D |

$$OR = \frac{\dfrac{A}{B}}{\dfrac{C}{D}}$$

**Fig 9. Odds ratio.** Calculation of odds ratio.

Based approach [20]. This approach assumes independence of features, so outlier detection is most appropriately applied after regressing out, where the features become more independent. Histogram-Based Outlier Score for an instance is calculated by building a histogram for each feature and aggregating them. The proportion of outliers or the amount of contamination of the dataset is a hyperparameter. In this task, for each plate containing wells, histogram-based outlier detection is applied separately with a 0.01 amount of contamination, and the outlier cells are removed. A useful library in Python for this section is PyOD [21]. A sensitivity analysis on the threshold of contamination is performed to assess stability of the results. As shown in Fig 10, it seems that our proposed method is robust to the slight changes of this hyperparameter.

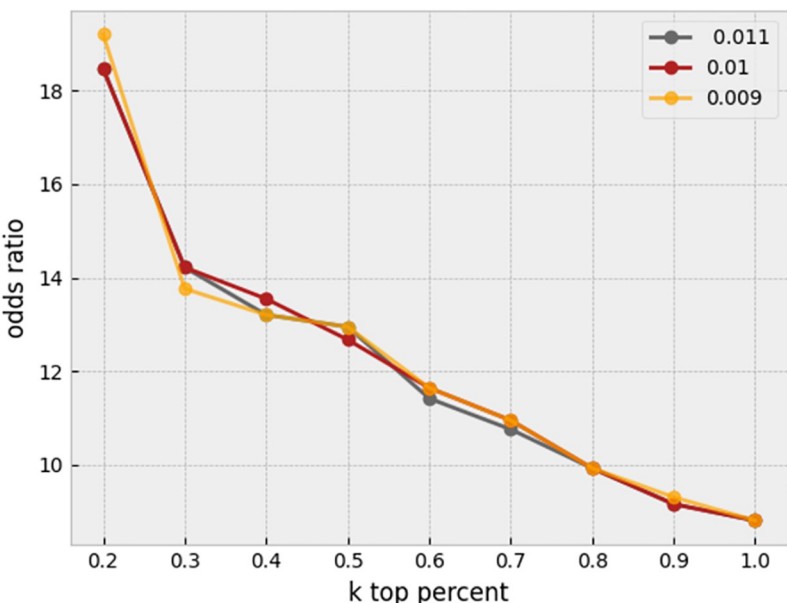

**Fig 10. Sensitivity analysis.** The proportion of outliers in each plate is set to different amounts of 0.009, 0.01, and 0.011 after applying preprocessing and before regressing out.

**Regressing-out.**  The majority of features are affected by a feature named *Cells_AreaShape_Area*. This feature expresses the space that is occupied by each cell. We used the regress-out technique to reduce the impact of this feature on the rest in order to strengthen the meaning of each feature individually. For this purpose, the relation between each feature and *Cells_AreaShape_Area* of all cells in one plate was modeled by fitting a linear equation without intercept. Considering x as *Cells_AreaShape_Area* feature of all single-cells in a plate and y as another feature from the feature set of all cells. In other words, $x_i$ represents the area and $y_i$ another feature of the $i^{th}$ cell in that plate. A linear regression line without intercept attempts to fit a line with an equation of the form $y = ax$. After fitting the model for each feature and cell area, that feature column was replaced by their residuals from the regression prediction. Residual in regression means the difference between any data point and the regression line:

$$Residual = real\ value - predicted\ value \qquad (1)$$

For each feature except cell area, and for each plate, this technique and replacement have to be done separately. By using this technique, significant progress was observed in the final results and odds ratio in different first k percentages.

**Validation methods.**  One important step in validating the results of the proposed method thoroughly is to design some experiments in order to assess the significance of the observed enhancements. We designed three experiments to this end.

*P-value assessment*. As explained in methodology section, the Fisher's exact test is used to test if highly correlated compound profiles have the same MoA. Considering the P-value of the Fisher's test can be used to determine how significant the results are.

*HBOS significance assessment*. If HBOS is a significant step in the proposed method, changing its hyperparameter should not change the overall trend of the odds ratio plot, and it should lie upon the base method.

*Per-drug validation*. In a drug discovery pipeline, there is an alternative strategy to get candidate compound matches. In this strategy, a compound with a known primary MoA is used as a query to find compounds that match its profile, and hopefully its primary MoA. Note that this setup is different from the earlier one in which only top correlated compound pairs in the entire experiment are considered as hits and are followed up. The difference stems from the fact that in the latter strategy of following up only strongest matches, the list of hits may involve only few compounds, while in the alternative strategy all compounds are involved in the test. Therefore, the alternative strategy is a more difficult test to pass. For the alternative strategy, Fisher's test is applied per drug; In other words, the contingency table is calculated for each drug to check if, for a specific compound, top k compounds with the highest correlation have the same MoA. Then, P-values of the Fisher's tests for all compounds are adjusted using the Bonferroni correction method, to account for doing multiple tests. The number of drugs with P-values smaller than various thresholds is counted for both base profiles after preprocessing and our proposed method for profile enhancement. Comparing these values can lead to an understanding of how significant the proposed method is.

**Deep learning.**  Recently, deep neural networks have appeared very promising in representation learning on various problems. Several methods were studied for the purpose of learning more meaningful representations to increase the similarity between drug embeddings with the same mechanism of action. Both supervised and unsupervised approaches are tested, and a brief explanation for each of them is discussed next.

*Classification with HSIC loss*. A simple model is designed for this task. The input of the network is single-cell features after feature selection and normalization, which was previously explained, and the output that the model has to predict is the drug by which that cell in the

input is treated with. We expect the network to learn a useful representation of cells through this auxiliary task of predicting drug identity from the cellular measurements. For preparing the training data, five mechanisms of action are chosen and about 30 drugs with these MoAs are selected randomly. All cells in various pairs of plates and wells that are treated by these 30 compounds are collected to form the training data. An important point to be careful in the train-test split is not to have cells from the same well/drug in both train and test/validation data.

The purpose of this random selection of compounds and not to use all of them in training and designing a network is to assess reasonable generalization on unseen data, especially new compounds whose mechanisms and similarities with others are still unknown. Let $(X, y)$ be a data point such that $X \in R^{470}$ and $y \in R^{30}$. As discussed earlier, each single-cell profile has about 470 features (less than 470), which are extracted by CellProfiler, and the output is probabilities for each of 30 compounds.

The loss function that is used for this task is Hilbert-Schmidt Independence Criterion (HSIC) loss [22] with the aim of learning on the training data with a limited number of MoAs and compounds but generalize and perform well on profiles that have not been seen by the model. This loss function has recently been proposed to facilitate out-of-distribution generalization. Few changes in the implementation of HSIC loss by authors are needed to use it for the multi-class classification.

The model is constructed by three dense layers. The activation function that is used for all layers is ReLU and after the last layer softmax was applied to the outputs. After training and convergence of loss on the train data, the trained model can be used for extracting new representations for every single cell. The second last layer or the hidden layer before the class probabilities (the red layer shown in Fig 11) is our new representation for the input single-cell profile.

After finding the representation of every single cell by passing them through the trained model, all previous steps such as aggregation, calculating the correlation between the compounds' profile, and plotting are repeated. The results from the new representations were meaningless, and were not promising compared to raw profiles, and did not improve the performance.

*Autoencoders*. Recently, Autoencoder [23] has achieved remarkable success as a feature extraction method. A basic autoencoder has two parts, an encoder, and a decoder. The output of the encoder represents the reduced representation of input data (code) and the decoder reconstructs the original input from the latent code that is generated by the encoder. Later, other variations of autoencoders were proposed that each tried to solve an issue such as variational autoencoder (VAE) [24], denoising autoencoder (DAE) [25], sparse autoencoder (SAE) [26], relational autoencoder [27], and etc. In this section, preparing the training data is quite similar to the classification task. Since this method is unsupervised, labels (compounds' names) are not needed. The decoder of the model should reconstruct the input of the encoder, which is a single-cell profile. After training and convergence of the mean squared loss on the training data, the trained model can be used for extracting new representations for every single cell. The code layer or the output of the encoder (the red layer shown in Fig 12) is the new representation for the input single-cell profile from the autoencoder.

After finding the representation of every single cell by passing them through the trained model, all previous steps for calculating the odds ratio are repeated. Basic AE, VAE, SAE, RAE, and DAE with different depths and code sizes were tested, but no significant improvement was observed in the final results. The convergence plot for SAE is included in S3 Fig.

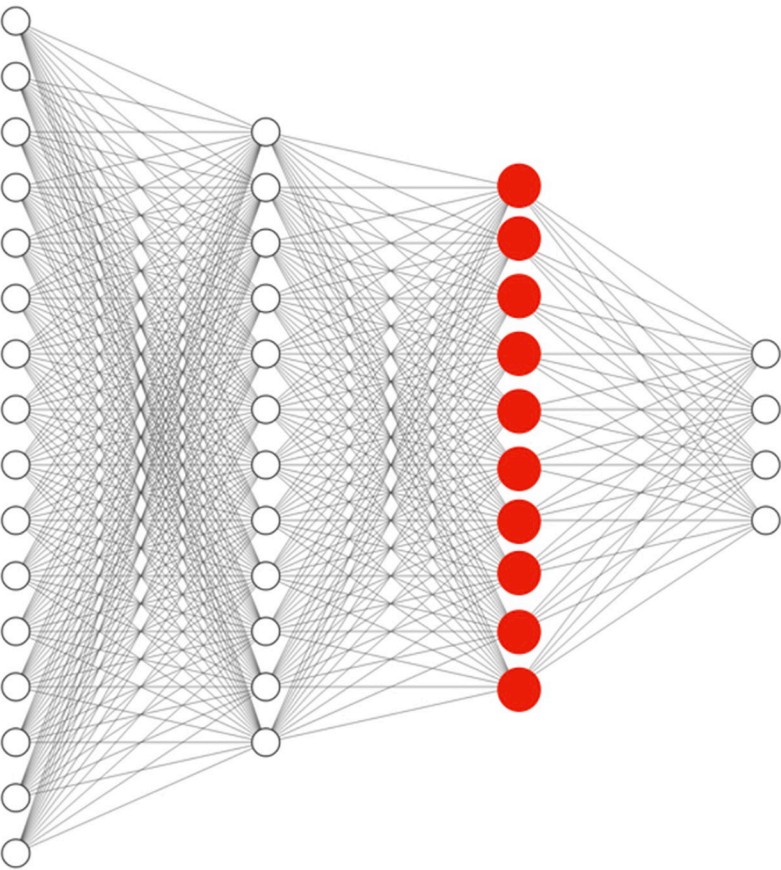

**Fig 11. Simple classification network architecture.** New representation from the classification task to feed into the pipeline is created by passing through the trained network and picking the red layer.

*Mixup.* According to [28], one way to improve model generalization is to construct some virtual training data:

$$\hat{x} = \lambda x_i + (1 - \lambda)x_j, \quad where\ x_i,\ x_j\ are\ raw\ input\ vectors. \tag{2}$$

$$\hat{y} = \lambda y_i + (1 - \lambda)y_j, \quad where\ y_i,\ y_j\ are\ onehot\ label\ encodings. \tag{3}$$

As proposed in the paper, mix-up is based on the prior knowledge that linear interpolations of feature vectors should lead to linear interpolations of the associated targets. This technique is used in the classification steps that were covered earlier in the HSIC section. It led to better accuracy on the test set of the auxiliary task, but no significant improvement in the odds ratio was observed.

## Experiments and results

### Data cleaning

The enrichment of k percentages of compound pairs with the highest profile correlation in having the same MoA is used for the evaluation. To measure the enrichment, the ratio of pairs with the same MoA to different MoAs is calculated, and compared to the same ratio for weakly

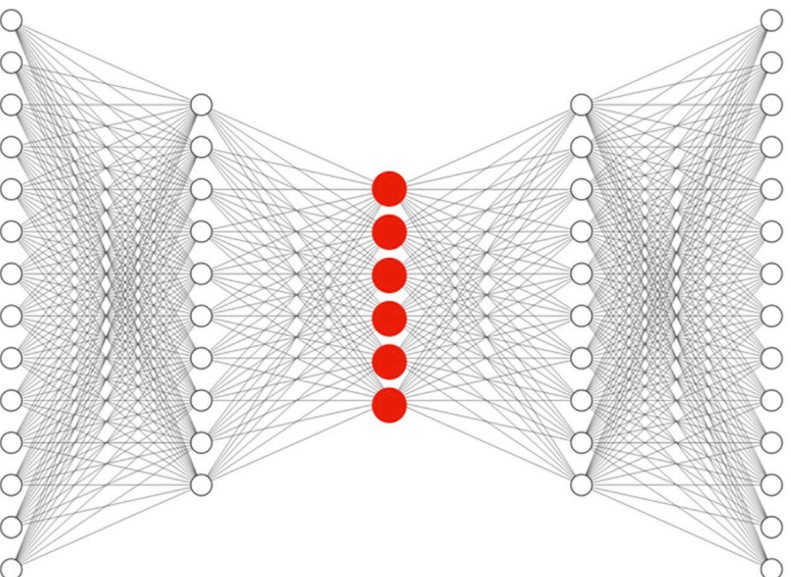

**Fig 12. Simple classification network architecture.** New representation from the classification task to feed into the pipeline is created by passing through the trained network and picking the red layer.

correlated pairs. This comparison could be summarized into a single number known as the odds ratio, which is described in the Material and Methods. This basically shows how many times we expect to find biologically related compound pairs in the top k percent connections compared to the other connections.

We first investigated a certain feature transformation step that aims to remove the cell area information that is implicitly contained in other features. This is particularly useful given that many morphological features, such as the perimeter, or texture, are affected by the cell area. We simply form a linear regression to predict any given feature from the cell area at the single-cell level. The residual, or error, of such a regression represents the information that is uncorrelated to the cell area. This process, which we call "regressing out," helps balance the effect of cell area in the profiles. Fig 13 left, shows that the odds ratio has improved slightly as a result of this step. We make comparisons in two cases. In the first case, all compounds in the dataset are

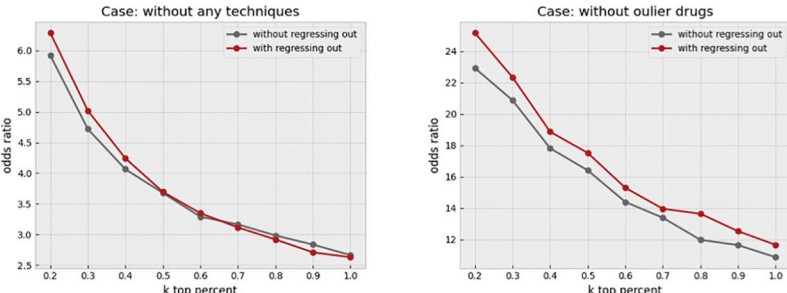

**Fig 13. Effect of regressing out the cell area.** The left plot shows the increase in the odds ratio by regressing out the cell area from all other features in different k percentages when no other data cleaning method is applied. The right plot shows that regressing out improves the result when just outlier drugs, which exhibit weak phenotypes, were removed.

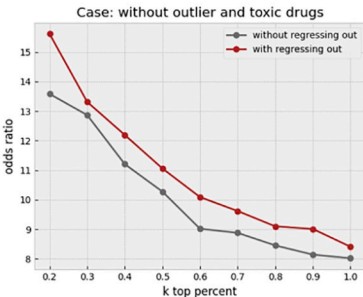
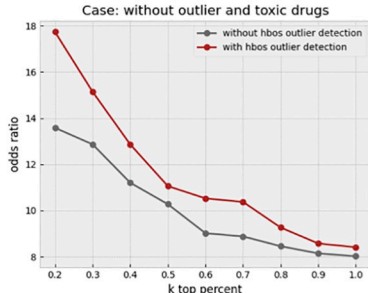

**Fig 14. Effect of regressing-out and cell outlier detection.** The left plot shows the increase in the odds ratio by regressing out the cell area from all other features in different k percentages when outlier and toxic drugs were removed. Improvement in the odds ratio becomes more significant when the toxic drugs are excluded from the analysis. The right plot shows that using histogram-based outlier detection improves the result in the same baseline case, where the outlier and toxic drugs were removed.

included, while in the second case, only the compounds with a high replicate correlation are retained. This filtering has previously been discussed to be a key step in removing treatments that do not show any strong biological effect from the lens of the image-based assay (4). We call such compounds "outlier" throughout the paper. It is notable that the odds ratio jumps to a much higher level, from 9 to higher than 40, when such compounds are removed. In this case, regressing cell area out results in a more consistent improvement in the odds ratio across various values of k, which is illustrated in Fig 13 right. We next investigated the effect of removing overly toxic compounds, which highly affect the cell viability. We would expect a much smaller cell count in the wells that are treated by such compounds. The profile might be of lower reliability as a result of the small cell count. In addition, the profiles of toxic compounds are so unique that artificially inflate the odds ratio. The increase in the odds ratio as a result of toxicity is not valuable from a practical perspective, due to limited clinical use of overly toxic compounds, and in drug discovery as a result. Therefore, we opted to remove such compounds before making further evaluations. Fig 14 left, shows that regress-out gives a more pronounced improvement over the odds ratio when both outlier and toxic compounds are removed. This could be partly due to regress-out unreliability under small sample size, which is the cell count in this case.

Finally, we applied outlier detection at the level of single-cells through a histogram-based technique, which is called "HBOS." Building upon our previous improvements, we further removed outlier cells prior to making the profiles. The odds ratio keeps enhancing in all the thresholds, as shown in Fig 14 right. It turns out that the cell outlier detection is a more effective method in improving the overall profile quality, while regress-out mainly improves the very top connections.

To assess the combined effect of regress-out and cell outlier removal, we take as baseline the non-toxic and inlier drugs in addition to one of these methods, and apply the other method next. Regress-out seems to provide an additional marginal improvement over cell outlier removal, Fig 15 left. In contrast, cell outlier detection gives more extra enhancement over regress-out, as shown in Fig 15 right. We conclude that both methods are valuable in cleaning the profile, and providing independent improvements, but the cell outlier removal is a bit more effective. All techniques have been described in more detail in the Material and Methods section.

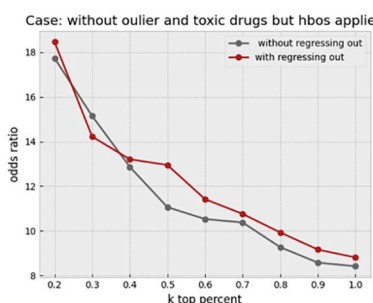
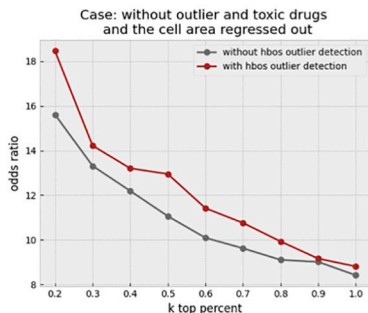

**Fig 15. The combined effect of outlier detection and regressing-out.** The left plot shows the effect of regressing out when all other preprocessing steps (outlier and toxic drug removal, and cell outlier removal) are applied. The right plot shows the effect of cell outlier removal when all other steps, including outlier and toxic drug removal and regress-out, have been applied. Cell outlier removal seems to provide a more consistent improvement across values of k compared to the regress-out technique.

## Validation results

We investigated three different experiments to validate the significance of odds ratio results.

Fig 16 shows the amount of log(P-value) of Fisher's tests, which their odds ratio can be found in the result section. We illustrated two situations: first, base profiles (only outlier and toxic drugs removed) and secondly, enhanced profiles (cell area regressed out from other features, and HBOS is applied). According to Fig 16, by tightening the condition, which means decreasing k, we have achieved a lower P-value, which indicates more confidence in both situations. The P-values of the proposed method are less than the base method, and it confirms the significance of our proposed method compared to the base method.

In another experiment, we tested the effectiveness of the HBOS outlier detection method with different hyperparameters. In all of the curves in Fig 17, all preprocessing steps are

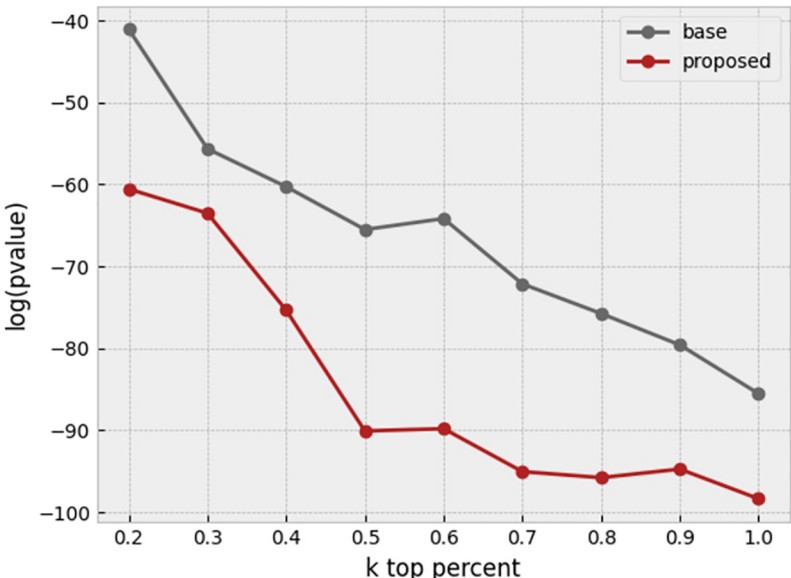

**Fig 16. P-value comparison between base profiles and our proposed enhanced profiles in different k percentages.**

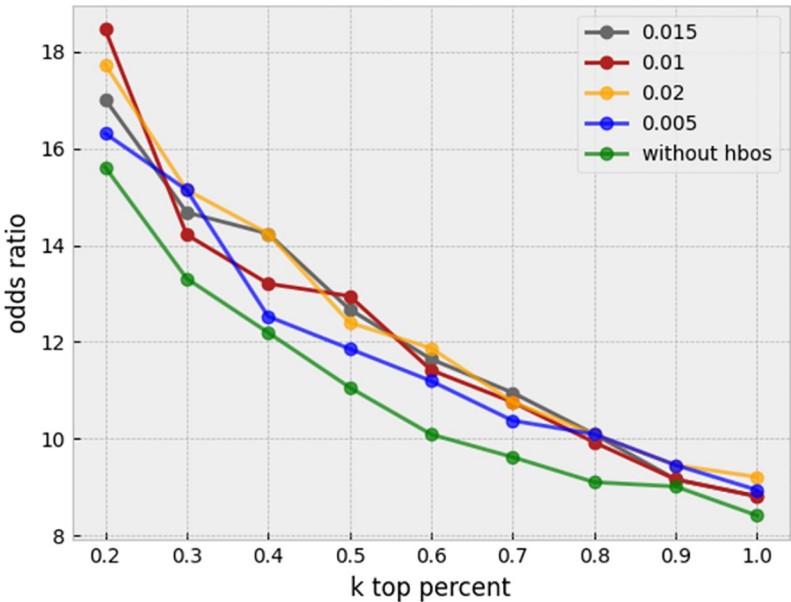

**Fig 17. HBOS significance analysis.** HBOS with different proportions of outlier removal has outperformed the case in which no outliers were removed.

applied to the profiles, but the HBOS contamination parameter is different among different curves. The green curve is related to profiles with no HBOS outlier detection. It is clear that HBOS with different hyperparameters has improved the profiles compared to not applying it.

In the last experiment, the effectiveness of the proposed method for profile enhancement is tested for each drug through the alternative strategy that was discussed in validation methods section. Here, we plot the number of queried compounds that their 10 top correlated compound matches mostly share the same MoA of the queried compound. In Fig 18 the plot is

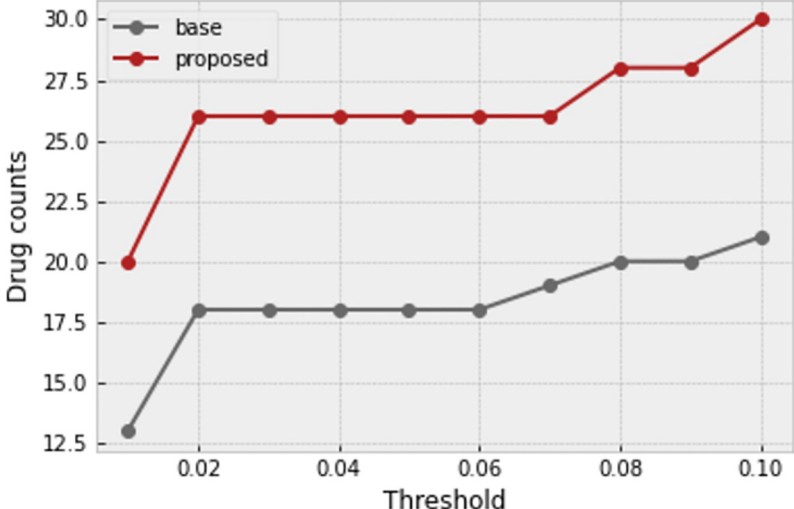

**Fig 18. Comparison of drug counts for each P-value threshold.** Drug counts for each P-value threshold are compared between base profiles and our proposed method (Thresholds in the plot are divided by the total number of drugs for using Bonferroni correction method in actual experiments).

made against various thresholds of the P-value that is used to decide the significance. As can be seen, the proposed method has improved the baseline significantly.

## Analysis of deep learning methods

Unfortunately, none of the deep representation learning methods, which are described in Material and Methods, could lead to a better result in the odds ratio (S1 and S2 Figs). There are some explanations for these observations. The most important factor that prevents representation learning, could be information loss, which is a result of using CellProfiler output. In fact, there are some important features that are captured using CellProfiler, but they are mostly the features that are introduced as important features by biologists, and there could be some hidden patterns in the raw images that a deep neural network is capable of learning, which are not captured by CellProfiler. For example, the texture information in the CellProfiler features are limited to certain Haralick texture descriptors, while a deep convolutional model could learn richer texture patterns. One other source of information loss in this work, is that the raw images contain 16 bits of information for each pixel. This means that they can be modeled more accurately when fed originally to a deep model, compared to the CellProfiler features, which are often of lower precision.

Another reason that the network does not improve the results is that there are some problems with outputs of CellProfiler for some images with low quality or with a small number of cells in them. This problem was explained in the previous sections. All these observations and hypotheses lead to one conclusion, that it is challenging for the neural networks to improve the CellProfiler output, and they should be trained on the raw images instead to be able to capture the high-quality information.

## Conclusion

High throughput assays play a vital role in carrying out biomedical experiments. CellProfiler is a common tool used to quantify the data produced by these assays, but many sources of error might affect the data quality in this pipeline. In this work, we mainly focused on data cleaning steps that could enhance image-based profiles extracted from CellProfiler.

Our Experiments indicate that data cleaning methods highly impact the quality of extracted features to identify mechanisms of action of different drugs, which results in a higher amount of the odds ratio. Removing drugs with intra-correlation less than inter-correlation improves the odds ratio. It is the result of keeping those drugs with more meaningful features. Removing toxic drugs—drugs that cause cell death—decreases the odds ratio, but makes the result more real and meaningful.

Due to the different error sources in the imaging and feature extracting pipeline, cell level outlier detection is an important step to enhance the profile of each specific compound. HBOS outlier detection was used in this work. Regressing out the cell area from all other features is another helpful step in data cleaning, since the cell area widely affects all other features, and by regressing it out, other useful information can be presented better in the resulting profile. Both techniques proved to be effectively improving the odds ratio.

Deep representation learning methods that are applied on top of CellProfiler features could not achieve a result better than the raw features, mainly because of information loss in the Cell-Profiler measurement step, but they have perfect ability in capturing hidden patterns, and they may be useful if applied on the raw images instead.

The proposed methods of this work are simply applicable to CellProfiler extracted features to improve the quality of image-based profiles, and can enhance the odds ratio significantly.

Therefore, they can be considered as a pivotal step in the profile pre-processing step of typical analysis workflows.

## Supporting information

**S1 Fig. Odds ratio for profiles that are obtained through representation learning using a denoising autoencoder.** A denoising autoencoder (DAE) with one hidden layer of encoder and decoder, and code size of 200 is trained. Compound-level profiles that are used for the last step come from the aggregation of DAE representation of cell measurements. Odds ratios in different percentages are too low and full of fluctuations compared to the cases that were investigated in the results section.
(TIF)

**S2 Fig. Odds ratio for profiles that are obtained through a supervised representation learning using the mixup technique.** A simple fully connected network that is regularized through applying the mixup technique is trained. In the first step, instead of raw cell-level profiles, representations that are extracted from the network are used. Considering this plot, the mixup technique does not effectively improve upon the baseline that was discussed earlier in the results section.
(TIF)

**S3 Fig. Convergence plot of loss in the train and validation sets during training sparse autoencoder to capture the representation of raw profiles.** This figure is a sample of convergence plots of deep methods in this work. Deep networks are tried for representation learning but they lead to no improvement in the odds ratio. It is clear that the loss has been decreased during the training process and no overfitting can be detected in the plots. This plot specifically represents sparse autoencoder training and validation losses.
(TIF)

## Author Contributions

**Conceptualization:** Mohammad Hossein Rohban.

**Data curation:** Arghavan Rezvani, Mahtab Bigverdi.

**Formal analysis:** Arghavan Rezvani, Mahtab Bigverdi.

**Investigation:** Arghavan Rezvani, Mahtab Bigverdi, Mohammad Hossein Rohban.

**Methodology:** Arghavan Rezvani, Mahtab Bigverdi, Mohammad Hossein Rohban.

**Project administration:** Mohammad Hossein Rohban.

**Resources:** Mohammad Hossein Rohban.

**Software:** Arghavan Rezvani, Mahtab Bigverdi.

**Supervision:** Mohammad Hossein Rohban.

**Validation:** Arghavan Rezvani, Mahtab Bigverdi, Mohammad Hossein Rohban.

**Visualization:** Arghavan Rezvani, Mahtab Bigverdi.

**Writing – original draft:** Arghavan Rezvani, Mahtab Bigverdi, Mohammad Hossein Rohban.

**Writing – review & editing:** Arghavan Rezvani, Mahtab Bigverdi, Mohammad Hossein Rohban.

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
