## [Decision Letter · Decision Letter 0]

10 Dec 2021

PONE-D-21-28958Data cleaning for image-based profiling enhancementPLOS ONE

Dear Dr. Rohban,

Thank you for submitting your manuscript to PLOS ONE. After careful consideration, we feel that it has merit but does not fully meet PLOS ONE’s publication criteria as it currently stands. Therefore, we invite you to submit a revised version of the manuscript that addresses the points raised during the review process.

ACADEMIC EDITOR:Based on the comments received from the reviewers and my own observation, I recommend major revisions for the article. Authors need not consider the references suggested by the reviewers if they are irrelevant or not required to strengthen the references.==============================

We look forward to receiving your revised manuscript.

Kind regards,

Thippa Reddy Gadekallu

Academic Editor

PLOS ONE

“NO authors have competing interests”

Reviewers' comments:

Reviewer's Responses to Questions

**Comments to the Author**

1. Is the manuscript technically sound, and do the data support the conclusions?

Reviewer #1: Yes

Reviewer #2: Yes

2. Has the statistical analysis been performed appropriately and rigorously? 

Reviewer #1: No

Reviewer #2: Yes

3. Have the authors made all data underlying the findings in their manuscript fully available?

Reviewer #1: Yes

Reviewer #2: Yes

4. Is the manuscript presented in an intelligible fashion and written in standard English?

Reviewer #1: Yes

Reviewer #2: No

5. Review Comments to the Author

Reviewer #1: A Data cleaning based on deep learning approach for image-based profiling enhancement is presented here. Topic is interesting. The manuscript is well-written. I would like to mention some comments to improve the quality of the manuscript.

-Abstract: please provide a conclusion sentence in its last sentence. The following sentence is not proper.\\

We also examined unsupervised and weakly-supervised deep learning based

methods to reduce the feature dimensionality, and finally suggest possible avenues for future research

-Introduction: I would like to see a comprehensive literature review about papers related to the topic.

-Section 2: It is recommended to add a methodology section

----Data used

----Methodology

------deep learning approach

------pre-processing

------validation

------ so on

-Section 3: please insert concepts in methodology section. Here you must mention results of the method.

-If the applied method has unknown parameters, I would like to see a sensitivity analysis. Please see the following paper.

A fuzzy-ga based decision making system for detecting damaged buildings from high-spatial resolution optical images

- Please insert convergence diagram of deep learning approach in the text.\\

-Validation regarding outputs should be done.

-The number of references is not adequate for a journal paper. Please mention at least 30 papers in the section.

Reviewer #2: The proposed research discusses a data cleaning approach for image-based profiling enhancement. It is a novel research area. However, the following concerns should be addressed.

• The title needs revision, and accordingly abstract and conclusion sections should also be updated.

• The results and discussions about how the proposed approach enhances the state of the art is missing. I recommend authors highlight the contribution of the proposed work separately, along with the limitations of the system.

• A detailed, layered design describing the proposed approach should be included for a better understanding of readers.

• The architecture looks very abstract and misses very important details. I recommend authors elaborate design and experimental setup of the proposed approach. The authors have described the materials and methods section, but I recommend including a detailed experimental setup for a better understanding and interpretation of the proposed work.

• The authors have included unnecessary references which can be removed and essential references such as, https://www.mdpi.com/1424-8220/20/24/7299”, “https://www.mdpi.com/1424-8220/20/20/5780” can be referred.

• I recommend authors to add a separate section to highlight novelties of the proposed.

Some more changes are needed:

4. All tables should be symmetrical and should follow a similar formatting style.

5. All the equations should be written using a professional equation editor and should use a similar formatting style and numbering.

6. Check the entire manuscript for grammatical and typo errors.

6. PLOS authors have the option to publish the peer review history of their article (what does this mean?). If published, this will include your full peer review and any attached files.

Reviewer #1: No

Reviewer #2: **Yes: **Sharnil Pandya

---

## [Author Response · Author response to Decision Letter 0]

1 Mar 2022

We thank the reviewers for their critical assessment of our work. In the following we address their concerns point by point.

Reviewer 1

Reviewer Point P 1.1 — Abstract: please provide a conclusion sentence in its last sentence. The following sentence is not proper: We also examined unsupervised and weakly-supervised deep learning based methods to reduce the feature dimensionality, and finally suggest possible avenues for future research

Reply: We removed the mentioned sentence of the abstract and inserted the following sentence as a conclusion sentence:

Our experiments indicate by performing these time-efficient preprocessing steps, image-based profiles can preserve more meaningful information compared to raw profiles. In the end, we also suggest possible avenues for future research.

Reviewer Point P 1.2 — Introduction: I would like to see a comprehensive literature review about papers related to the topic

Reply: We investigated several papers related to the topic, specifically those related to cell quality control and outlier detection. A summary of these papers is inserted into the introduction section.

Reviewer Point P 1.3 — Section 2: It is recommended to add a methodology section • Data used

• Methodology

– deep learning approach – pre-processing

– validation

– so on

Section 3: please insert concepts in methodology section. Here you must mention results of the method.

Reply: We agree with the reviewer on this important point. The ”Materials and methods” section is updated as follows:

• Dataset

• Methodology

 – Preprocessing

 – Removing outlier drugs

 – Removing Toxic drugs

 – Evaluation metric

 – Histogram-based outlier score 

 – Regressing-out

 – Validation methods

 ∗ P-value assessment

 ∗ HBOS significance assessment ∗ Per-drug validation

 – Deep learning

 ∗ Classification with HSIC loss

 ∗ Autoencoders 

 ∗ Mixup

We also changed the order of sections ”Materials and methods” and ”Experiments and results”. Now, ”Materials and methods” come prior to ”Experiments and results”.

Reviewer Point P 1.4 — If the applied method has unknown parameters, I would like to see a sensitivity analysis.

Reply: The only unknown hyperparameter in the applied method is HBOS contamination paramater. The sensitivity analysis of this parameter is added to ”Histogram-based outlier score ” subsection in methodology section.

Reviewer Point P 1.5 — Please insert convergence diagram of deep learning approach in the text

Reply: Convergence diagram of sparse autoencoder is added to the supplementary materials. Reviewer Point P 1.6 — Validation regarding outputs should be done.

Reply: For validation, we designed three experiments. The description of experiments is under ”val- idation methods” section in materials and methods, and the result of validation experiments is under ”validation results” in experiments and results section. In one of the experiments we investigated the p-value in the fisher test for different k values. During this investigation, we found out that the p-value of k=0.1 is not as significant as other k values. Therefore, we updated all of the figures and removed k=0.1 from all comparisons.

Reviewer Point P 1.7 — The number of references is not adequate for a journal paper. Please mention at least 30 papers in the section

Reply: This specific area is not studied in many papers, therefore the number of references was under 30 at first. But during updating the literature review section, we increased the number of references.

Reviewer 2

Reviewer Point P 2.1 — The title needs revision, and accordingly abstract and conclusion sections should also be updated.

Reply: We changed the title to ”Image-based cell profiling enhancement via data cleaning methods”. We also updated the abstract and substituted the last sentence with another sentence that indicates our contribution. We highlighted our novelty in both introduction and conclusion sections, too.

Reviewer Point P 2.2 — The results and discussions about how the proposed approach enhances the state of the art is missing. I recommend authors highlight the contribution of the proposed work separately, along with the limitations of the system.

Reply: We changed the first figure of the paper, and we think it highlights the contribution more effectively. Furthermore, we changed several sentences in the introduction and conclusion sections so that the enhancement of our method would be more evident.

Reviewer Point P 2.3 — A detailed, layered design describing the proposed approach should be included for a better understanding of readers

Reply: We change the first figure of the paper in the introduction section to make our approach more understandable and clear to the reader.

Reviewer Point P 2.4 — The architecture looks very abstract and misses very important details. I recommend authors elaborate design and experimental setup of the proposed approach. The authors have described the materials and methods section, but I recommend including a detailed experimental setup for a better understanding and interpretation of the proposed work.

Reply: We added the methodology section in materials and methods to describe the steps more accurately. In each step, hyperparameters and even libraries are included.

Reviewer Point P 2.5 — I recommend authors to add a separate section to highlight novelties of the proposed

Reply: According to previous concerns, we highlighted our novelty in different parts: abstract, intro- duction, and conclusion. Furthermore, the new figure 1 shows the novelty more clear and understandable to readers.

Reviewer Point P 2.6 — 1- All tables should be symmetrical and should follow a similar format- ting style. 2- All the equations should be written using a professional equation editor and should use a similar formatting style and numbering. 3- Check the entire manuscript for grammatical and typo errors.

Reply: All done.

---

## [Decision Letter · Decision Letter 1]

6 Apr 2022

Image-based cell profiling enhancement via data cleaning methods

PONE-D-21-28958R1

Dear Dr. Rohban,

We’re pleased to inform you that your manuscript has been judged scientifically suitable for publication and will be formally accepted for publication once it meets all outstanding technical requirements.

Kind regards,

Thippa Reddy Gadekallu

Academic Editor

PLOS ONE

Additional Editor Comments (optional):

Reviewers' comments:

Reviewer's Responses to Questions

**Comments to the Author**

1. If the authors have adequately addressed your comments raised in a previous round of review and you feel that this manuscript is now acceptable for publication, you may indicate that here to bypass the “Comments to the Author” section, enter your conflict of interest statement in the “Confidential to Editor” section, and submit your "Accept" recommendation.

Reviewer #1: All comments have been addressed

2. Is the manuscript technically sound, and do the data support the conclusions?

Reviewer #1: Yes

3. Has the statistical analysis been performed appropriately and rigorously? 

Reviewer #1: Yes

4. Have the authors made all data underlying the findings in their manuscript fully available?

Reviewer #1: No

5. Is the manuscript presented in an intelligible fashion and written in standard English?

Reviewer #1: Yes

6. Review Comments to the Author

Reviewer #1: The authors answered the questions carefully. The manuscript is mature for the publication. Please consider it for publication.

7. PLOS authors have the option to publish the peer review history of their article (what does this mean?). If published, this will include your full peer review and any attached files.

Reviewer #1: No

---

## [Editor Report · Acceptance letter]

25 Apr 2022

PONE-D-21-28958R1 

Image-based cell profiling enhancement via data cleaning methods 

Dear Dr. Rohban:

I'm pleased to inform you that your manuscript has been deemed suitable for publication in PLOS ONE. Congratulations! Your manuscript is now with our production department. 

Kind regards, 

on behalf of

Dr. Thippa Reddy Gadekallu 

Academic Editor

PLOS ONE